# Fast Exact Unlearning for In-Context Learning Data for LLMs

**Andrei Muresanu** [1 2]   **Anvith Thudi** [3 2]   **Michael R. Zhang** [3 2]   **Nicolas Papernot** [3 2]

## Abstract

Modern machine learning models are expensive to train, and there is a growing concern about the challenge of retroactively removing specific training data. Achieving exact unlearning in deep learning pipelines—producing models as if certain data had never been included in training—remains an open problem. In this paper, we revisit exact unlearning in deep learning and show that for large language models (LLMs) we can efficiently exactly unlearn "fine-tuning data" (the data used to adapt a pre-trained model). This follows from two observations. First, we can use in-context learning to adapt the LLM to the fine-tuning dataset instead of SGD based algorithms. Second, we show that accurate in-context learning can be done with quantized k-means, which allows for effectively constant time unlearning operations. Our evaluation shows that this unlearning recipe has similar performance to fine-tuning alternatives, but vastly reduces the unlearning costs. Our study also highlights the need for new measures of unlearning cost when adapting the learning algorithm to have faster unlearn operations.

## 1. Introduction

After a machine learning model is deployed, it may become necessary to deploy a new model that does not use part of the original training set. This can occur because of legislation on the "right to be forgotten" (Mantelero, 2013) or because of unknown data provenance (e.g., some data may have come from untrustworthy sources). Machine unlearning addresses this challenge of modifying a model to behave as if it were trained without including certain datapoints. Specifically, exact unlearning aims to produce the same model (distribution) the learning algorithm would produce if trained without a set of datapoints.

Efficient exact unlearning for deep learning remains an open problem. Advances in exact unlearning have come from identifying cheap operations to simulate what changing the dataset does to learning. This has been possible for classical machine learning algorithms (Cao & Yang, 2015) (Brophy & Lowd, 2021), such as clustering (Ginart et al., 2019). However, simulating the change caused by removing a datapoint from an SGD training run with a neural network (deep learning) has proven difficult, with current approaches still requiring cost on the order of training the original model (Bourtoule et al., 2021).

Approximate unlearning has been proposed as an alternative to exact learning and aims to lower unlearning costs by approximating the distribution of models trained without the desired datapoints. However, there is currently no consensus on evaluation metrics, making comparison and utility difficult to evaluate (Thudi et al., 2022a; Hayes et al., 2025)

In this paper, we show that exact unlearning can be efficient for "fine-tuning" a pre-trained large language model (LLM), i.e., learning given access to an LLM. The past literature on exact unlearning had not considered learning algorithms that leverage pre-trained models. However, this has become a recent trend in private machine learning (Yu et al., 2021; De et al., 2022), where fine-tuning is easier to learn privately. We show that a similar trend also holds for exact unlearning.

In-context learning with LLMs offers an alternative to traditional model fine-tuning that proves amenable to exact unlearning. In-context learning works by selecting representative examples from a training dataset to guide the LLM's responses. While in-context learning achieves comparable performance to weight fine-tuning on many downstream tasks (Brown et al., 2020), it presents a distinct challenge for unlearning: we must reproduce the example selection process as if certain datapoints never existed in the fine-tuning dataset.[1] This is non-trivial because effective example selection depends on understanding the complex relationships between datapoints.

We observe that accurate in-context learning algorithms

---

[1]Department of Computer Science, University of Waterloo, Waterloo, Canada [2]Vector Institute, Toronto, Canada [3]Department of Computer Science, University of Toronto, Toronto, Canada. Correspondence to: Andrei Muresanu <andrei.muresanu@uwaterloo.ca>.

*Proceedings of the $42^{nd}$ International Conference on Machine Learning*, Vancouver, Canada. PMLR 267, 2025. Copyright 2025 by the author(s).

---

[1]Throughout this paper we consider the pre-training and fine-tuning datasets to be independent.

reduce to clustering over some feature space, which can admit efficient exact unlearning. Specifically, a common in-context learning algorithm is to run k-means on embeddings of training examples (Zhang et al., 2022). Instead, we propose to run quantized k-means (Ginart et al., 2019), which admits unlearning operations independent of the dataset size. With this, we can unlearn the fine-tuning data independent of model and dataset size.

We introduce ERASE, an unlearning approach that combines in-context learning with quantized k-means clustering. Our empirical evaluations explored how it compared to existing exact unlearning baselines. We conducted experiments across Big-Bench Instruction Induction (BBII) (Zhou et al., 2023) tasks, and compared performance of ERASE to variants of SISA (Bourtoule et al., 2021) (an optimized exact unlearning algorithm for SGD-based learning). We found that for most tasks we evaluated, ERASE matched or improved the accuracy of SISA, and in all cases had drastically cheaper unlearning operations.

Finally, in studying fast exact unlearning, we make the observation that existing "faster" exact unlearning algorithms for deep learning also increase inference cost. To the best of our knowledge, this was not discussed in past work, and we present an analysis of how many inference passes per unlearning request a method can handle after which it is less efficient than the baseline of retraining.

To summarize, our contributions are:

1. Showing that for certain datasets, exact unlearning can be efficiently achieved by using in-context learning.

2. Proposing an exact unlearning algorithm, ERASE, for in-context learning, that has dataset and model-size independent unlearning operation costs.

3. Identifying the trade-off between inference cost and unlearning cost and proposing a new holistic cost metric. We use this metric to study when in-context learning provides more efficient unlearning deployments as compared to fine-tuning alternatives such as SISA.

## 2. Background

### 2.1. Unlearning

Exact machine unlearning asks, given a training algorithm returning some model (parameters, cluster centroids, etc.) $T : D \rightarrow M$ and a datapoint $x$ in the training dataset $D$, to return the output of $T(D \setminus x)$ given an output from $T(D)$ (Cao & Yang, 2015); if $T$ is random, then this means returning a sample according to its model distribution. The "trivial" solution for unlearning is to retrain on $D \setminus x$, i.e., run $T(D \setminus x)$ without using the model $T(D)$. A goal in

developing unlearning methods is when machine unlearning can be achieved cheaper than this baseline (i.e., when having access to $T(D)$ makes undoing $x$ fast). Towards this goal, past work has shown faster machine unlearning algorithms for a variety of classical machine learning algorithms (Cao & Yang, 2015; Ginart et al., 2019; Brophy & Lowd, 2021). For deep learning, known methods for exact unlearning partition data before training such that retraining can be computed more efficiently (Bourtoule et al., 2021), though the cost is still on the order of training (and can come with performance degradation). Given the difficulty of exact unlearning, past work has considered various notions of approximate unlearning, and methods for these (Thudi et al., 2022a; Golatkar et al., 2020; Graves et al., 2021; Baumhauer et al., 2022; Guo et al., 2019; Thudi et al.; Pawelczyk et al., 2023; Yao et al., 2023; Jang et al., 2022; LLMS; Chen & Yang, 2023). However a consensus for metrics to use for approximate unlearning has not yet been reached. For example, there was a Neurips competition on standardizing metrics (Triantafillou et al., 2024). Moreover, it is not clear if approximate notions of unlearning are suitable for some applications of unlearning, such as legal requirements. Our work avoids this ambiguity by focusing on exact machine unlearning, and showing access to a pre-trained model opens ways to learn with faster exact unlearning operations.

A separate body of literature related to unlearning (what is considered in this paper) focuses on knowledge unlearning where the goal is to remove an undesired "behaviour" of an LLM as opposed to unlearning specific datapoints. We refer the reader to Cooper et al. (2024) for a detailed comparison of these two distinct bodies of literature.

### 2.2. In-Context Learning

Large language models are able to learn to perform a task and make predictions for new inputs by conditioning on a few input-label pairs (*few-shot examples*), with no fine-tuning. This is known as in-context learning and was observed to be an emergent behaviour in GPT-3, which is a language model with 175B parameters trained on Internet-scale data (Brown et al., 2020). Subsequent work on in-context learning has studied why in-context learning works (Min et al., 2022), including viewing the language model as implicitly performing Bayesian inference (Xie et al., 2021). The effectiveness of in-context learning may vary significantly with the choice of in-context demonstrations (Lu et al., 2022). For instance, Zhang et al. (2022) highlighted the role of diversity in the examples chosen for performance and proposed methods such as a baseline of random sampling examples, or clustering the examples using kmean++ and including the example closest to the centroid in each cluster. This last approach is called Auto Chain-of-Thought (ACoT). In our work, we show the in-context learning framework has faster exact unlearning operations.

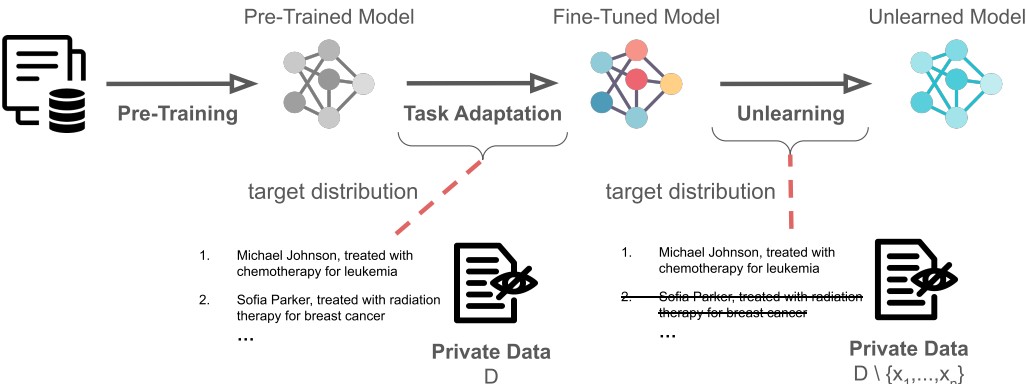

*Figure 1.* An illustration of our threat model for unlearning requests in the fine-tuning stage of training. Sensitive data can be introduced at the fine-tuning stage of a neural network and may need to be unlearned. We assume this sensitive data does not overlap with the pre-training dataset, meaning no change to the pre-trained model is needed to unlearn. In particular, while unlearning pre-training data efficiently is an open problem, we show unlearning in this second stage can have performative and efficient exact unlearning methods. Model trainers may consider moving data that may need to be unlearnt to the fine-tuning stage for efficient unlearning.

# 3. Taxonomy: Exact Unlearning Methods for Fine-Tuning LLMs

Chowdhury et al. (2025) is the only previous work we are aware of studying unlearning in the fine-tuning stage (i.e., task adaptation stage) of LLMs. However, their approach does not constitute exact unlearning; see Appendix F for details. Despite this, we note that various retraining-based approaches to exactly unlearn are already applicable. We now describe existing methods to exactly unlearn in the task adaptation stage, noting these can be classified along what the task adaptation learning algorithm is (that they are unlearning for). In describing these baseline methods, we focus on contrasting unlearning and inference cost.

**Measuring Cost**    We follow the online model of Ginart et al. (2019) for unlearning and cost. We consider a stream of unlearning requests $x_1, \cdots, x_i$. For each request we must produce the output of $T$ (the training algorithm) if it was just given $D \setminus \{x_1\}, D \setminus \{x_1, x_2\}, \cdots D \setminus \{x_1, \cdots, x_i\}$ where $D$ was the training set. We measure unlearning cost as the average cost over a uniformly sampled i.i.d stream of unlearning requests from the training dataset. When unlearning one point, the unlearning cost is the expected cost to unlearn a point when uniformly sampling from the dataset. We note that requests may not always be uniformly sampled (Gupta et al., 2021), and that manipulation to the order of datapoints can lead to increased compute costs (Shumailov et al., 2021), but do not consider these adversarial settings in this paper.

## 3.1. Parameter Fine-tuning

The standard exact unlearning approach for fine-tuning with SGD-based algorithms is Sharded, Isolated, Sliced, and Aggregated training (SISA) (Bourtoule et al., 2021). $n$-SISA constructs an ensemble of $n$ models trained on disjoint equal sized partitions of the dataset $D$ (effectively training each model for $1/n^{\text{th}}$ the cost of the original model), called sharding. The second stage of SISA is slicing, which limits a data point to being used in a specific interval of training, e.g., the first 3 epochs. However, as seen in Figure 4, our models converge after one epoch of SISA fine-tuning. Hence, in this paper, we assume each model trains on a data point only once and do not further use SISA slicing. To unlearn with SISA, only one of the shorter training runs needs to be redone (which had the datapoint to unlearn), and in expectation half of that training run needs to be redone, giving an asymptotic unlearning rate of $O(\frac{1}{n} \times$ cost to train one model fully on D). This gives a $1/n$ factor improvement over retraining naively, but it is known that by increasing $n$ one decreases the performance of the ensemble model returned by SISA (Bourtoule et al., 2021): there is a limit to how large $n$ can be. Intuitively, each of the models sees less and less data as $n$ grows, eventually giving an ensemble of noisy predictions.

While the unlearning operation cost decreases with $n$, the inference cost for $n$-SISA grows with $n$. The inference cost is $O(n * f(t))$ where $t$ is the number of tokens in the inference question and $f$ represents the cost of inference per token for one model. For a Transformer with a constant attention window (as in our experiments), the cost is $O(n * t^2)$ and we set $f(t) = O(t^2)$ for the rest of the paper.

## 3.2. In-context Learning

As an alternative to parameter fine-tuning for task adaptation, in-context learning algorithms output a set of $k$ examples from the fine-tuning dataset $D$ which are then prepended to the input given to an LLM. In-context learning

makes no modifications to the parameters, and the only dependence on the dataset $D$ is the set of examples to prepend. Hence, to unlearn in-context learning, we need to (re)sample examples from the distribution of examples the in-context learning algorithm would output if given just $D \setminus \{x^*\}$. As no past work has proposed unlearning methods for in-context learning, we overview past in-context learning algorithms and the cost of naively retraining for unlearning.

A baseline in-context learning method is to randomly sample in-context examples (Zhang et al., 2022) from the task adaptation dataset. Hence the distribution of samples for unlearning $x^*$ will just be the distribution (sampled without replacement) of $k$ uniform samples from $D \setminus x^*$. We can do this cheaply by rejection sampling. We resample once from $D \setminus \{x_1, \cdots, x_k\}$ to replace $x^*$ if it was sampled as an in-context examples ($x^* \in \{x_1, \cdots, x_k\}$), and do nothing if $x^*$ was not sampled (as the conditional that $x^* \notin x_1, \cdots, x_k$ is the same for both $D$ and $D \setminus x^*$). Hence this gives a constant $O(1)$ unlearning cost per example, giving $O(m)$ for unlearning $m$ examples.

For certain tasks, a more effective in-context learning algorithm is Auto Chain-of-Thought (Zhang et al., 2022) (ACoT), which takes embeddings of the examples in $D$ and produces $k$ clusters from them using k-means++. ACoT then picks the example from each cluster that is closest to the centroid. This is often combined with prompting the model for an explanation for its response, but we ignore this step for this paper (and note this does not need changing for unlearning). Hence, ACoT is effectively k-means++ on the in-context examples, and unlearning would require producing the cluster centroids k-means++ would return if run on $D \setminus \{x^*\}$. We are unaware of efficient unlearning methods for generic k-means++, and hence consider naively retraining as the unlearning method. Note running k-means++ has a cost $O(|D|d)$ when fixing the number of iterations, where $d$ is the dimension of the embedded in-context examples. Hence, assuming negligible change to the dataset size over $m$ unlearning requests, we have the unlearning cost for ACoT over $m$ unlearning requests is $O(m|D|d)$. We also highlight that simply resampling the removed in-context example from the k-means++ centroid clusters is not sufficient for exact unlearning because the selection of the removed example is not independent to the rest of the training set. Therefore, retraining (rerunning the clustering algorithm) or an unlearnable alternative to k-means++ is required.

These in-context learning methods have unlearning operation costs independent of model size, in contrast to SISA; however ACoT's unlearning operation scales with dataset size. For all in-context learning methods, the prepending of $k$ in-context examples increases the cost to run inference. The inference cost is $O(t^2 + k \times s)$, where $s$ is the average number of tokens in an in-context example, and one has to

---

**Algorithm 1** In-context Learning with ERASE

**Require:** A set of training examples $D$, the desired number of in-context examples $k$, and quantization parameter $\epsilon$
**Ensure:** Examples $q^{(i)} = [q_1^{(i)}, q_2^{(i)}, \ldots q_k^{(i)}]$ for in-context learning

1: **for** each example $q$ in $D$ **do**
2:     Encode $q$ with feature extractor e.g., Sentence-BERT
3: **end for**
4: Cluster all the encoded example representations into $k$ clusters using quantized k-means with quantization parameter $\epsilon$
5: **for** each cluster $i = 1, \ldots, k$ **do**
6:     Sort examples $q^{(i)} = [q_1^{(i)}, q_2^{(i)}, \ldots]$ in the ascending order of the $\ell_2$ distance to the quantized cluster centroid
7: **end for**
8: Return $q_1^{(i)}$ for $i = 1, \ldots, k$

---

run a forward pass over $k$ of them alongside the original test example of $t$ tokens. This is in contrast with $n$-SISA which only scaled as $O(n \times t^2)$, and note $s$ is typically longer than the normal input (see example format in Section 5.1).

To summarize these different costs, we present the different unlearning operation and inference costs in Table 1 (which includes the in-context learning algorithm we will propose), where we now see a spectrum of increasingly more efficient unlearning algorithms: no model dependence for all in-context learning methods and furthermore no dataset dependence with random sampling and our algorithm. At the same time, we see inference costs get higher as we make unlearning operations more efficient.

## 4. Methodology

Given the current takeaways from the brief taxonomy provided in Section 3, we now proceed to propose a new algorithm to fill a gap in unlearning operation efficiency. Furthermore, given the inverse trends observed between inference cost and unlearning cost, we propose a new holistic measure of unlearning costs to capture how frequent unlearning requests must be for an algorithm to be more efficient: this being particularly useful to compare unlearning cost when changing the learning algorithm.

### 4.1. ERASE : A New In-context Learning Method Designed for Unlearning

The discussion in Section 3.2 reveals a gap for an in-context learning algorithm that is as accurate as ACoT, but with unlearning operation costs independent to the dataset size. We propose such an algorithm in Algorithm 1, which we call ERASE (Efficient Removal And Selection of Examples).

| Cost | $n$-SISA | ACoT | ERASE | Random |
|---|---|---|---|---|
| Unlearning Op | $O(\frac{m}{n}\text{Train}(D,\text{model}))$ | $O(m\lvert D\rvert d)$ | $O((m^2 d^{5/2}/\epsilon)$ | $O(m)$ |
| Inference Op | $O(n \times t^2)$ | $O(t^2 + k \times s)$ | $O(t^2 + k \times s)$ | $O(t^2 + k \times s)$ |

*Table 1.* Asymptotic unlearning and inference costs of different exact unlearning methods at the task adaptation stage of an LLM. Costs are presented for unlearning $m$ data points, where $D$ is the dataset, $d$ is the dimension of the embeddings used for the in-context examples, and $\epsilon$ is the quantization parameter in ERASE . Here $m$ is assumed to be $O(1)$ asymptotically as $D$ grows. Here $k$ is the number of in-context examples, $t$ is the number of tokens in the input question, and $s$ is the average number of tokens in an in-context example. We do not consider model size for inference as that does not affect relative comparisons: would introduce the same constant for all methods.

ERASE builds on past work that highlighted the role of selecting diverse examples for accurate in-context learning (Zhang et al., 2022). ERASE takes the set of training examples $D$ and computes the Sentence-Bert embeddings for each $q \in D$. It then creates $k$ clusters from the embeddings (the desired number of in-context examples) and orders the examples in ascending $\ell_2$ distance to the centroid: i.e., for each cluster $i \in [k]$ returns a list $q^{(i)} = [q_1^{(i)}, q_2^{(i)}, \cdots]$. ERASE then returns $q_1^i \ \forall i \in [k]$, the examples closest to each centroid. This algorithm is similar to ACoT (Zhang et al., 2022). However, ERASE uses quantized k-means (Ginart et al., 2019) to cluster instead of k-means++.

Quantized k-means allows for dataset size independent unlearning of clusters (which is the main cost for unlearning the selection of in-context examples). The algorithm computes cluster centroids similar to k-means, but adds an additional imbalance correction, and most importantly, quantizes the centroids to a randomly sampled $\epsilon$-lattice. More formally, quantized k-means quantizes the intermediate centroids $c_i \rightarrow \hat{c}_i \in \epsilon * \mathbb{Z}^d + \theta$ for some $\theta \in Unif([-1/2, 1/2]^d)$ and quantization parameter $\epsilon$. By making $\epsilon$ larger we make the final cluster centroids more stable to changes (would require a big shift in the un-quantized centroid to make a shift after quantization). It was shown that with high probability $1 - O(1/\epsilon\lvert D\rvert)$ over uniformly sampling a datapoint to unlearn (Ginart et al., 2019), removing that datapoint does not change the final clusters and hence unlearning can be achieved by doing nothing.

In particular, the unlearning operation cost of quantized k-means for $m$ unlearning requests is $O(m^2 d^{5/2}/\epsilon)$ in expectation where $d$ is the dimension of the embedding vectors, as stated in Theorem 4.1 in Ginart et al. (2019). The proof follows by the previously mentioned bound on the probability a deletion request changes the centroids (which introduces the $1/(\epsilon\lvert D\rvert)$ dependence), and further noting the cost to check this stability for a given datapoint is independent of the dataset size. Finally, the $(1/\lvert D\rvert)$ probability of having to retrain cancels with the $\lvert D\rvert$ factor involved in retraining from scratch, giving the stated dataset independent cost for the unlearning operations.

However, quantized k-means introduced a quantization parameter $\epsilon$ which one would need to be tuned for the task. For the tasks we evaluated on, we found a fixed value of $\epsilon = 0.05$ led to comparable accuracy to ACoT.

### 4.2. Holistic Unlearning Costs: Towards A More Nuanced View of Benefits to Unlearning Efficiency

As previously noted in Table 1, methods that decrease unlearning operation cost also increase inference cost. To the best of our knowledge, the literature has not previously discussed this trade-off, and how it can make the baseline of finetuning with 1-SISA cheaper if there are a large number of inference queries per unlearning request. We propose to evaluate when exactly various unlearning algorithms are more efficient than the baseline of 1-SISA (i.e., fine-tuning and and doing naive retraining to unlearn) This comparison is motivated by previous deployment comparisons of in-context learning and fine-tuning (Liu et al., 2022). This metric is stated in the following definition:

**Definition 4.1** (Holistic Unlearning Cost). We compute the holistic unlearning cost for a method $M$ with unlearning operation cost $U_M$ and inference cost $I_M$, as the number of inferences per unlearning request upon which $M$ costs the same as 1-SISA. That is, the solution to

$$U_M + I_M * C(M) = U_{\text{1-SISA}} + I_{\text{1-SISA}} * C(M)$$

where $C(M)$ is the number of inferences. This gives

$$C(M) = (U_M - U_{\text{1-SISA}})/(I_{\text{1-SISA}} - I_M)$$

## 5. Experiments

We now empirically explore the benefits of unlearning given by in-context learning, compared to fine-tuning. First, we explore the test accuracy of our method ERASE compared to other in-context learning methods, and find it performs on par with ACoT (but recall has dataset-independent unlearning costs). We hence take ERASE to represent in-context learning and compare it to the unlearning baseline for task adaptation of fine-tuning with SISA (Bourtoule et al., 2021). We find it has comparable test accuracy across many tasks, and by carefully picking the number of in-context examples, can give significant benefits to holistic unlearning costs (i.e.,

| Method | WinoWhy | Timedial | Sports Understanding | Logical Fallacy Detection |
|--------|---------|----------|----------------------|---------------------------|
| 4-SISA | $1.4 \times 10^3$ | $0.2 \times 10^2$ | $2.7 \times 10^3$ | $1.3 \times 10^3$ |
| **2-shot** | $\mathbf{2.5 \times 10^3}$ | $\mathbf{0.4 \times 10^2}$ | $\mathbf{5.2 \times 10^3}$ | $\mathbf{2.2 \times 10^3}$ |
| 3-shot | $1.7 \times 10^3$ | $0.3 \times 10^2$ | $3.4 \times 10^3$ | $1.5 \times 10^3$ |
| 4-shot | $1.2 \times 10^3$ | $0.2 \times 10^2$ | $2.6 \times 10^3$ | $1.1 \times 10^3$ |

*Table 2.* Number of inferences per unlearning request to be as expensive as 1-SISA: higher is better. These values are computed using the inference and unlearning operation costs for the methods reported in Tables 3 and 4 (Appendix B) respectively. We see that using 4 in-context examples is more expensive than 0-shot 4-SISA, as it can handle fewer inferences per unlearning request. This is despite it having more efficient unlearning operations. Nevertheless, in-context learning with 2-shot is still better for holistic unlearning costs than parameter finetuning (with a factor of improvement of $\approx 2$ across the board) and by Figure 3 has comparable performance on the tasks.

how infrequent unlearning can be for it to be better than 1-SISA).

In summary, ERASE performs on par with ACoT, and doing task adaptation (i.e., "fine-tuning") using ERASE is more cost-effective for unlearning than SISA.

### 5.1. Experimental Setup

LLMs now commonly use a causal decoder architecture (Zhao et al., 2023). Within this architecture class, we observe similar performance across pre-training procedures. Therefore, we use the popular LLaMA (Touvron et al., 2023) as a representative base model within all experiments. We evaluated all methods on a suite of 15 tasks. All experiments were run on a single node containing four A40 Nvidia GPUs.

**Task Selection**   The 15 tasks we evaluate on are from Big-Bench (Srivastava et al., 2023) (released under the Apache 2.0 license). Big-Bench tasks are designed to be difficult for LLMs to solve. From Big-Bench we selected tasks to emphasize challenging scenarios and isolate in-context learning ability. Our task selection process started with considering only tasks from the Big-Bench Instruction Induction (BBII) (Zhou et al., 2023) subset of Big-Bench, which are curated to be difficult and lack any instructional content within examples. We then filter out any BBII tasks with less than 200 examples, to ensure each task has adequate examples for fine-tuning and evaluation. We chose to exclude the Dyck Languages task from BBII because its format is prohibitive toward log probability based evaluations. To have a more comprehensive dataset, we supplement these with 5 additional tasks from the broader Big-Bench dataset which the untuned version of LLaMA solves with no higher than a 40% normalized aggregate score. Finally, we remove the "task prefix" from the prompts of all tasks. This step helps isolate in-context learning ability from confounding variables such as instruction following ability.

**Fine-Tuning Setup**   We fine-tune all model parameters using a pipeline based on Alpa (Zheng et al., 2022). We use a loss mask to update gradients based only on the log

probabilities of answer tokens. We mask all tokens within the example input and task setup (e.g., the tokens for "Input:" and "Output:"), fine-tuning only on the token within the answer string. We use a block size of 256 tokens and batch size of 8. We use the Adam optimizer (Kingma & Ba, 2017) with $\beta_1 = 0.9$, $\beta_2 = 0.98$, weight decay of 0.01, and learning rate of 1e-5. We also use 10 warm-up steps with a linear schedule. The full list of training parameters can be found in Table E.

**Hyperparameter Selection**   We found that the learning rate was the only hyperparameter with a significant impact on final performance after fine-tuning. To tune our learning rate, we fine-tune our model on the intent recognition dataset for the rates: {5e-5, 1e-5, 5e-6, 1e-6, 5e-7, 1e-7} and choose the one with the lowest test perplexity (1e-5). Finally, we try three different number of warm-up iterations: 10, 15, and 40 steps, and find that 10 warm-up steps performed best.

**Prompt Formatting**   We describe in Appendix A how we prompted an LLM for inference during: fine-tuning, zero-shot inference, and few-shot inference. This formatting is relevant to how we measured inference cost.

**Evaluation Setup**   Our evaluation of performance follows the process outlined by the BigBench dataset (Srivastava et al., 2023). All BigBench tasks we used are JSON tasks. JSON tasks have two types of evaluation metrics: generative and multiple-choice. Given an input, generative metrics require the model to generate some text. This text is then compared to a predefined correct answer with either no formatting or very little formatting. Multiple-choice metrics evaluate the model's output on a small set of multiple-choice options. The metric then either assigns the most likely choice as the model's answer or looks at the difference between the model's output distribution and the target distribution. We use the metric(s) recommended for the task, giving preference to using log-probability based multiple-choice metrics because continuous evaluation metrics provide smoother and more predictable changes in model performance (Schaeffer et al., 2023). We report the performance on each task

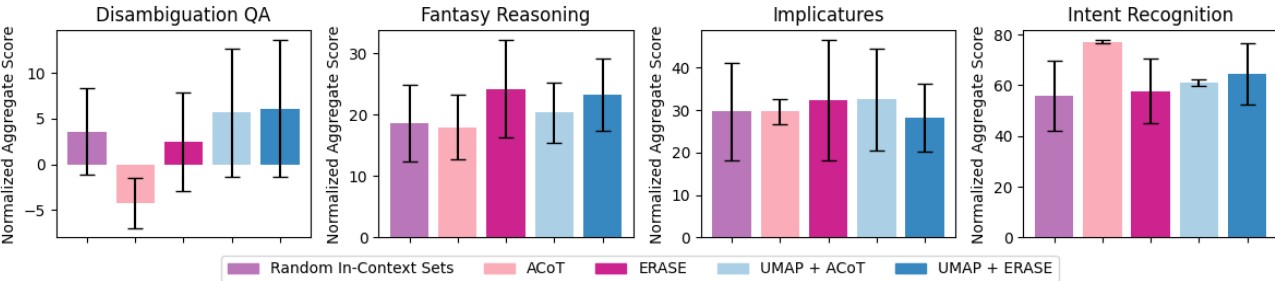

*Figure 2.* Comparison of the normalized aggregate score on 4 bigbench tasks between random selection of in-context examples, ACoT, and ERASE alongside dimension reduction variant of ERASE and ACoT (using UMAP). All methods are tested in the 4-shot setting. We see that ERASE matches our outperforms ACoT on three of the four tasks, and similarly with random selection. Considering dimension reduction for ERASE , we observed it made slight improvements but did not affect the relative improvement of ERASE over ACoT (still better than dimension reduced ACoT on three of the four tasks).

by the model's normalized aggregate score, which is normalized such that 0 represents random performance and 100 represents the performance of a human expert. The definition of normalized aggregate score (Srivastava et al., 2023) is described in Appendix A

### 5.2. ERASE  is comparable to Auto Chain-of-Thought

In this section we ask, does making in-context learning more efficient to unlearn affect its test performance? Here we compare ERASE to the other in-context learning algorithms described in Section 3 which were not optimized for unlearning. In particular, we investigate how ERASE compares to ACoT when ACoT outperforms random sampling (i.e. example selection is important), enabling dataset independent unlearning operation costs on those tasks.

We compared ERASE to ACoT and random selection of in-context sets in Figure 2, using default parameters of 10 iterations for clustering and setting the quantization parameter for quantized k-means to $\epsilon = 0.05$. We found that ACoT and ERASE outperform random selection on most tasks, i.e., the more expensive to unlearn methods perform better. Amongst these methods, we found that ERASE outpeforms ACoT on three out of the four tasks. Hence we conclude that ERASE provides and alternative to ACoT.

We further evaluated dimension reduction variants (using UMAP) of the clustering methods to understand whether embedding dimension affects the comparison between ACoT and ERASE . Considering the effect of dimension reduction, we see it slightly boosts (or matches) the performance for ERASE and ACoT. However, both ACoT and ERASE still perform comparably across the 4 tasks after dimension reduction. We conclude that dimension reduction has minimal relative impact on performance between ACoT and ERASE, hence not affecting the previous claim of comparability in accuracy between the two methods.

To conclude, we observe that ERASE has comparable accuracy to ACoT, despite having more efficient unlearning operations. We proceed to compare the accuracy of ERASE against the finetuning baselines.

### 5.3. Few Shot ERASE is comparable to SISA variants

The question we now ask is: can ERASE perform as well as the SISA variants (the baseline for fine-tuning with unlearning), and hence lead to alternative task adaptation algorithm with efficient unlearning operations for these tasks? To do this, we first compare the performance of ERASE with a different number of in-context examples to SISA with a different number of shards (i.e., models used for ensembling). Recall, increasing the number of in-context examples is expected to increase performance of ERASE , while decreasing unlearning operation cost and increasing inference cost. However, increasing the number of shards for SISA is known to reduce performance (Bourtoule et al., 2021), but also reduce the unlearning operation cost while increasing inference cost (Bourtoule et al., 2021). The comparisons across these ablations will later help us have a more fine-grained understanding of when in-context learning has a better holistic unlearning cost versus performance trade-off.

Specifically, we consider SISA with ensembles of size 4, 2, and 1.[2] In all implementations of SISA, we also do not use slicing as we train for only a single epoch and hence every datapoint is only used once (and thus already implicitly sliced). For ERASE , we use upto 4 in-context examples, and keep the same hyperparameters as in our previous comparisons between in-context learning methods.

---

[2]We do not test beyond 4 models per ensemble as we want each model in the ensemble to fine-tune on at least 20 examples for all tasks. This is as Figure 4 in Appendix A shows convergence in normalized aggregate score only ever occurs after 20 examples, where here we tested different sizes of ensembles.

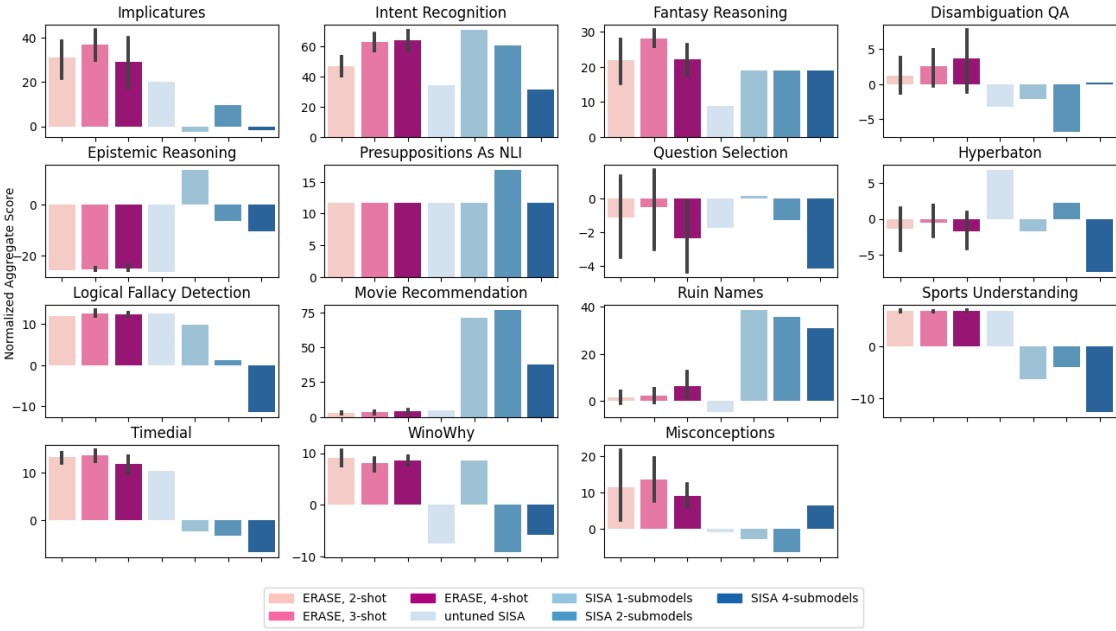

*Figure 3.* A comparison of the performance (measured by normalized aggregate score, defined in Section 5.1) of 2, 3, 4-shot ERASE to 1, 2, 4-SISA, and the baseline of no task adaptation for 15 bigbench tasks. We see on several tasks that ERASE performs comparably or even better than the SISA variants; LLaMA is capable of in-context learning with ERASE on these tasks. We repeat experiments with ERASE 10 times to estimate the standard deviation and evaluate all methods on the entire test set.

In Figure 3, we compared the test accuracy of the methods. We see that ERASE using 3 or 4 in-context examples can be as accurate as all the SISA variants for more than half of the 15 tasks, with 2-shot ERASE often still being as accurate as the most efficient to unlearn SISA variants. In short, ERASE performs comparably to SISA on many tasks.

### 5.4. Few Shot In-context Learning is holistically more cost efficient to unlearn than SISA

With this understanding of the predictive performance of ERASE relative to SISA, we turn to when it is cheaper to deploy in-context learning than SISA in the presence of unlearning requests. Specifically, how infrequent can unlearning requests be for each method after which they are worse than 1-SISA in cost, and does in-context learning allow more settings?

Towards answering this, we first reported the numerical cost (measured in FLOPS) per unlearning operation and per inference for the different methods in Table 3 and Table 4 (in Appendix B) respectively. This is reported for 4 tasks where in-context learning performs as well as SISA, as shown in Figure 3. In particular, we computed the expected cost per unlearning operation for SISA by taking the flops used to train an individual model in the ensemble and dividing by 2 (as dividing by 2 captures the expected length of training that needs to be redone if one logs checkpoints). Compared to the costs of SISA, which do not scale favourably to LLMs

as they are model size dependent, the model and dataset-independent unlearning costs of in-context methods are practically 0. For inference costs, we use the Flops Profiler package (Li, 2023), measuring base LLaMA to have an inference cost of roughly $264, 996, 864 \times (1 + \text{context length})$ FLOPS and evaluated the expected context lengths for all methods.

Given these values, we then used Definition 4.1 to compute the holistic unlearning costs for the different methods. We report the found number of inferences per unlearning request after which a method is more expensive than the baseline (1-SISA) in Table 2. Inspecting the values, We found that the best-performing version of in-context learning (the 4-shot version) costs more than the SISA variants to unlearn; that is, they can handle fewer inferences per unlearning request before being more expensive than the baseline. This is despite it having a more efficient unlearning operation.

However, note 2 and 3-shot ERASE still performed comparably to the most efficient to unlearn SISA variants. Hence we compared their holistic unlearning cost to 4-SISA (note that 4-SISA is more efficient to deploy than 3 and 2-SISA by Definition 4.1). We observed 3-shot ERASE has costs comparable to 4-SISA, and 2-shot ERASE in fact significantly improves the holistic unlearning costs: it allows unlearning requests to be at least almost half as infrequent as what 4-SISA allows before being as expensive as 1-SISA.

We hence conclude that in-context learning can be more

favourable for deployments subject to unlearning requests at the task adaptation phase. However, this requires picking the number of in-context examples more carefully and our trade-offs may not generalize to other tasks (where more in-context examples are needed for performance).

## 6. Conclusion

We showed that fast exact unlearning is possible for fine-tuning an LLM (i.e., task adaptation). This followed by using in-context learning for fine-tuning, and noting quantized k-means on embeddings is an accurate in-context learning algorithm. This provided a learning plus unlearning recipe whose unlearning operations costs were independent of both the LLM and dataset size. This efficiency gain was true even for a holistic unlearning cost that studied how frequent inference requests can be per unlearning request.

We note in-context learning is not always an effective learning method, and focused our study on situations where in-context learning is performant. For example, in-context learning can help a model understand the format of a trivia task, but can not significantly help a model recall more trivia. Additionally, in-context learning is limited by the context size and does not modify model weights. Therefore, it is not useful for learning over large amounts of data. We leave these ill-suited tasks, as well as an analysis of the trade-off between fine-tuning ability and unlearning cost, as future work.

## Impact Statement

This paper presents work whose goal is to advance the field of Machine Learning. There are many potential societal consequences of our work, none which we feel must be specifically highlighted here.

## Acknowledgements

We thank Eleni Triantafillou and Jamie Hayes for helpful comments on earlier versions of this work. We would like to acknowledge our sponsors, who support our research with financial and in-kind contributions: Amazon, Apple, CIFAR through the Canada CIFAR AI Chair, DARPA through the GARD project, Intel, Meta, NSERC through the Discovery Grant, the Ontario Early Researcher Award, and the Sloan Foundation. Anvith Thudi is supported by a Vanier Fellowship from the Natural Sciences and Engineering Research Council of Canada. Michael Zhang is supported by the NSERC CGS Fellowship. Resources used in preparing this research were provided, in part, by the Province of Ontario, the Government of Canada through CIFAR, and companies sponsoring the Vector Institute.

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

# A. Experimental Details

**Prompt Formatting**    Below we list how we prompted an LLM for inference during: fine-tuning, zero-shot inference, and few-shot inference. This formatting is relevant to how we measured inference cost. For completeness, we also describe the expected output during fine-tuning.

1. **Fine-tuning** *Input:* "[test example input]" *Output:* "[test example output]"

2. **Zero-Shot inference** *Input:* "[test example input]"

3. **Few-Shot inference** *Input:* "Input: [in-context example 1 input] Output: [in-context example 1 output] Input: [in-context example 2 input] Output: [in-context example 2 output] ... Input: [in-context example k input] Output: [in-context example k output] Input: [example input]"

**Normalized Aggregate Score**    We state the formula as defined in Srivastava et al. (2023) below, where for a given task "raw preferred metric" is the model's performance using the task's preferred evaluation metric, "low score" is the lowest possible score (typically random choice), and "high score" is the highest possible score (typically expert human performance):

$$[\text{normalized aggregate score}] \quad = \quad [\text{normalized preferred metric}] \quad = \quad 100 \times \frac{[\text{raw preferred metric}] - [\text{low score}]}{[\text{high score}] - [\text{low score}]}$$

# B. Additional Tables and Figures

| Method | WinoWhy | Timedial | Sports Understanding | Logical Fallacy Detection |
|---|---|---|---|---|
| 1-SISA | $71 \times 10^{12}$ | $70 \times 10^{12}$ | $71 \times 10^{12}$ | $70 \times 10^{12}$ |
| 4-SISA | $14 \times 10^{12}$ | $14 \times 10^{12}$ | $14 \times 10^{12}$ | $14 \times 10^{12}$ |
| **In-context Methods** | $\approx \mathbf{0}$ | $\approx \mathbf{0}$ | $\approx \mathbf{0}$ | $\approx \mathbf{0}$ |

*Table 3.* Unlearning operation costs of different exact unlearning methods, measured in FLOPS, for four different tasks: lower is better. All tasks have roughly the same unlearning operation cost for a given SISA method despite having dramatically different input lengths. This is as the examples for all the tasks fit entirely within the model's context window. Therefore, because fine-tuning requires no token generation, it means that all examples require only a single forward pass to compute the required fine-tuning log probabilities regardless of length. When compared to the costs of the SISA methods, we have the unlearning costs for in-context methods are $\approx 0$ given they are model size independent; notably random in-context sets are $O(1)$ (just the cost of resampling).

| Method | WinoWhy | Timedial | Sports Understanding | Logical Fallacy Detection |
|---|---|---|---|---|
| **1-SISA** | $\mathbf{13.5 \times 10^9}$ | $\mathbf{78.6 \times 10^9}$ | $\mathbf{7.0 \times 10^9}$ | $\mathbf{14.9 \times 10^9}$ |
| 4-SISA | $54.2 \times 10^9$ | $314.4 \times 10^9$ | $27.9 \times 10^9$ | $59.7 \times 10^9$ |
| 2-shot | $42.1 \times 10^9$ | $238.1 \times 10^9$ | $20.7 \times 10^9$ | $46.1 \times 10^9$ |
| 3-shot | $55.5 \times 10^9$ | $323.7 \times 10^9$ | $27.6 \times 10^9$ | $61.5 \times 10^9$ |
| 4-shot | $70.4 \times 10^9$ | $395.5 \times 10^9$ | $34.5 \times 10^9$ | $76.7 \times 10^9$ |

*Table 4.* Inference costs of different exact unlearning methods, measured in FLOPS, for four different tasks: lower is better. Unlike what was seen with unlearning operation costs in Table 3, we see a variety of inference costs depending on the task. Nevertheless, comparing costs across the SISA methods and in-context methods, we see that using 4 in-context examples has the largest cost. 3-shot in-context learning has inference cost comparable to 4-SISA, with 2-shot being the cheapest before the baseline of 1-SISA. We calculated the FLOPS for inference reported for inference by computing the inference cost per token for the model, the average number of tokens in an input, and the average number of token in an in-context example for each task.

# C. Discussion

Here we discuss how future work can further explore unlearning different parts of the deep learning pipeline (Section C.1), as was found successful in this paper. We also discuss how work on the limits to holistic unlearning efficiency (as introduced in

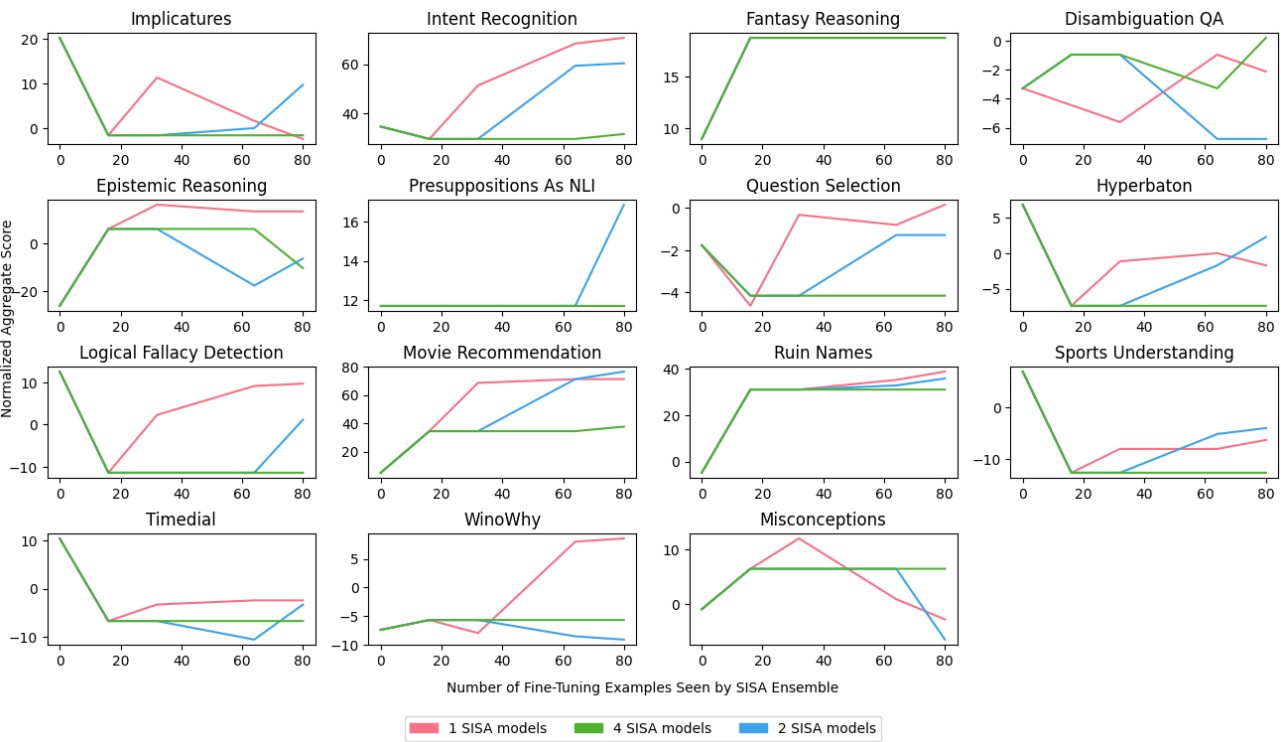

*Figure 4.* Performance (reported by normalized aggregate score) of SISA variants as they progress through training, measured by the number of training examples seen. This is shown for the 15 bigbench tasks selected according to the process described in Section 5.1, using the finetuning setup also described in Section 5.1. We find that finetuning with SISA typically converges after 80 examples, and always required at least 20 examples to converge. Hence we use 20 examples to define a hard cut-off on how small our shards can be for SISA. Recall normalized aggregate score reports model performance such that 0 represents random performance and 100 represents the performance of human experts.

this paper) has implication to the more general trustworthy ML community (Section C.2). Lastly, we describe how changing the unlearning definitions could change cost comparisons (Section C.3), and the open problem of understanding when in-context learning is better than fine-tuning (Section C.4) which has implication to the applicability of our observations.

### C.1. Towards More Fine-Grained Unlearning Threat Models

One core contribution of our work is the insight that the deep learning pipeline can be segmented, allowing for more efficient exact unlearning in specific stages. Practically, this suggests data with potential unlearning requirements should be allocated to stages where exact unlearning is more feasible. In this paper we split the deep learning pipeline into two stages (pretraining and task adaptation), as illustrated in Figure 1.

However this is still a coarse-grained view of a deep learning pipeline. Many large models go through multiple stages of refinement, intentionally using different data at each stage (Dubey et al., 2024). This can be at the pretraining stage where more carefully curated data mixtures are introduced after training on less filtered data to improve performance. Alternatively, in the task adaptation stage, a variety of fine-tuning techniques on different sources of data may be used before finally doing in-context learning, in order to do task adaptation for multiple tasks.

As an example of another unlearning problem arising from these modern pipelines, many large models run a hyperparameter selection over the mixture weights of large datasets by evaluating performance of models trained with those mixtures on smaller benchmark data (Dubey et al., 2024; Blakeney et al., 2024). Essentially they had learnt mixture weights of "public data" with this benchmark data. One can imagine this benchmark data contains sensitive user data reflecting specific user groups the model trainer is trying to improve performance for. Hence, how might one unlearn the mixture weights if some of the users in the benchmark data revoked access to their data? Is there a preferred method for mixture finding if one would

need to unlearn?

Our research aims to inspire future investigations into various stages of modern deep learning pipelines to assess the unlearning requirements for data at each stage and develop efficient unlearning methods. As our understanding evolves, we may identify efficient unlearning techniques for multiple pipeline stages, allowing strategic placement of potentially unlearnable data, e.g., in placement for retrieval based models as in (Min et al.). This approach could ultimately lead to efficient exact unlearning solutions for deep learning by addressing unlearning challenges in specific, critical parts of the process rather than tackling the entire pipeline simultaneously.

### C.2. On Lower-Bounds to Holistic Unlearning Cost and Implications to Trustworthy Machine Learning

Our investigation into exact unlearning methods for task adaptation revealed unlearning methods that decreased the costs for the unlearning operation often also increased the costs for inference. To quantify this trade-off, we proposed a new holistic measure of unlearning cost: the frequency threshold of unlearning requests at which a method becomes more cost-effective than retraining. This metric offers potential insights into the fundamental limits of exact unlearning efficiency.

Further study into this trade-off may be able to provide provable, or even empirically measured, lower-bounds on how holistically efficient unlearning can be in certain settings while still having an accurate learning algorithm. Our experiments suggest a possible unavoidable trade-off between inference cost and unlearning operation cost in certain machine learning setups, although more efficient algorithms may yet be discovered. Given that our algorithms achieve near-constant unlearning operation costs, future improvements would likely focus on reducing inference costs, which we expect can only improve by constant factors.

Such lower bounds would have implications beyond just unlearning. In particular, a lower-bound on holistic unlearning cost would imply a trade-off between how much data any accurate learning algorithm can be "invariant" to (i.e., can unlearn by doing nothing) and how efficiently such invariance can be checked. This is as one can construct an unlearning algorithm by checking if removing a training datapoint changes the final model and if not, do nothing; if significantly many points do not affect the model and this can be checked quickly, this provides a fast unlearning operation. This algorithm would also not increase inference cost, providing an example of a good holistic unlearning algorithm which such a lower-bound would prevent. Hence, lower-bounds to holistic unlearning cost may be of fundamental interest to the core machine learning community.

Moreover, a variant of the unlearning algorithm sketched above was proposed by Thudi et al. (2024), where checking invariance was possible with a new per-instance differential privacy analysis. However, such an analysis is currently slow to run, and a lower-bound on holisitc unlearning cost may explain the limits of how efficiently such per-instance DP analysis can be run. Hence, such a lower-bound to holistic unlearning can be of interest to the privacy community.

Given these connections, we believe understanding the limits of holistic unlearning to be of fundamental interest to the trustworthy machine learning community in general. Beyond any other connections, it may help illuminate why efficient exact unlearning for deep learning has been illusive.

### C.3. On Unlearning Definitions and Implications to Cost

We wish to reiterate here the distinction between exact and approximate unlearning, and the issues that are still common to both and how this can affect claims of efficiency. Approximate unlearning attempts to emulate exact unlearning in a manner that is sufficient for specific goals of unlearning. However, the reasons to unlearn may be several-fold in practice (legal, performance, etc) and not amenable to any specific metric, hence difficult for approximate unlearning. Nevertheless, for the cases where approximate unlearning is appropriate, it may be that there are similar trade-offs between inference cost and unlearning operation cost as was found for exact unlearning in this paper, or between other sources of deployment cost. Perhaps these trade-offs are also affected by the choice of approximate unlearning metric, giving an argument for or against certain metrics. In short, we are not aware of work looking into the efficiency trade-off between unlearning for different approximate unlearning metrics, and what holistic consideration (analogous to changing inference cost in exact unlearning) would need to be made.

We also note that exact unlearning is also not immune to problems of applicability. In particular, exact unlearning can still suffer issues of not being auditable (Thudi et al., 2022b), meaning additional requirements are potentially needed to satisfy legal requirements. It may be that to make these unlearning methods auditable, we may find a new trade-off between various sources of deployment costs, such as communication cost vs. offline cost for cryptographic protocols.

Furthermore, exact unlearning can also still leak privacy if not done with additional privacy requirements (Gupta et al., 2021), e.g., if an adversary gains access to the model before and after the unlearning operation for a data point they may be able to infer what the data point to unlearn was. It may be that satisfying privacy requirements on unlearning can change the ratios between inferences and unlearning requests we observed (for a given test accuracy), and the preference for in-context learning. For example, privacy often comes with accuracy degradation. However, the mechanism needed to make unlearning in-context examples private may involve less degradation to accuracy than for fine-tuning, and vice-versa.

Lastly, regardless of whether we are doing exact or approximate unlearning, note we assumed a stream of individual independent unlearning requests, but if they were aggregated into batches, then 1-SISA unlearning operation cost does not scale with the number of requests making it more desirable. Analogously, if the sequence of unlearning requests were not independent, further algorithmic changes may be needed to do unlearning (Gupta et al., 2021).

To summarize, we believe careful consderation of the specific unlearning requirements can affect our observed claims of which unlearning methods are more efficient. We hope future work considers the different settings described here, adding to our understanding of when unlearning can be efficient.

### C.4. Understanding when In-Context Learning Performs Best

Our experiments, as illustrated in Figure 3, showed that ERASE often achieves comparable accuracy to SISA. However, this was not always the case, and we observed a large variance in the relative performance between tasks. Furthermore, we are not aware of work that predicts when in-context learning is accurate (relative to fine-tuning) and we do not have an explanation for our observed variance in competitiveness.

We hypothesize that this variability is largely influenced by the pre-trained model, and that different pre-trained models may yield different results for in-context learning performance. This uncertainty has practical implications for unlearning: to determine if in-context learning can be used effectively (for more efficient learning) without significant performance loss compared to fine-tuning, it appears necessary to still fine-tune the model for comparison. Given that in-context learning is now known to be important for unlearning, we believe an important open problem for the trustworthy machine learning community is predicting when in-context learning is suitable for a task without requiring fine-tuning for comparison. Solving this problem would significantly enhance our ability to implement efficient unlearning strategies across various tasks and models.

## D. Statistical Significance Analysis

In this section, we analyze the statistical significance of our results from Figure 2. We analyze the results individually for each task and avoid cross-task analysis as with a sample size of 4, the analysis would not be meaningful. For our analysis, we assume that the performance of each algorithm is normally distributed with unknown population mean and variance. We also assume that the variance of each algorithm is independent. This assumption is based on the observation that each algorithm's variance is different and the intuition that some algorithms are more stable than others. We will conduct our analysis in a pairwise fashion between the following algorithms: Random In-Context Sets (Random), ACoT, and ERASE, which we will sometimes note as $A_1, A_2, A_3$. Formalizing the above, for $i \in \{1, 2, 3\}$ we assume the performance of each $A_i$ is sampled from some normal distribution $N(\mu_i, \sigma_i^2)$. Our Null Hypothesis is that each pair of two algorithms $A_i, A_j$, $i \neq j$ have equal mean performance, i.e., $\mu_i = \mu_j$. We test the Null Hypothesis using a two-sample t-test with assumed unequal variances. Table 5 contains the results of Figure 2, and Table 6 shows the calculated p-values.

## E. List of Training Parameters

A complete list of all training parameters used can be found in Table E.

## F. Exact Unlearning Commentary Of Chowdhury et al.

Chowdhury et al. (2025) propose unlearning by reverting to a checkpoint that did not observe the datapoint to be unlearned. However, this does not reproduce the model that would have been obtained had the datapoint never been included in that slice (as the model distribution is now dependent on where the datapoint to unlearn originally appeared). Furthermore, their method of selecting among models trained on permuted data slices deviates from the a priori distribution of models without the datapoint, and is conceptually analogous to the forging attack described by Thudi et al. (2022b). More generally, their

*Table 5.* Tabular format results from Figure 2. Each experiment (task-algorithm combination) had 32 independent trials. As described in Figure 2, ERASE matches or outperforms ACoT on most tasks, and similarly with random selection.

| Task | Algorithm | Mean | Std Dev |
| --- | --- | --- | --- |
| Disambiguation QA | Random In-Context Sets | 3.59 | 4.70 |
| | ACoT | -4.22 | 2.78 |
| | ERASE | 2.48 | 5.40 |
| | UMAP + ACoT | 5.65 | 6.99 |
| | UMAP + ERASE | 6.11 | 7.53 |
| Fantasy Reasoning | Random In-Context Sets | 18.69 | 6.26 |
| | ACoT | 18.02 | 5.21 |
| | ERASE | 24.24 | 7.93 |
| | UMAP + ACoT | 20.40 | 4.92 |
| | UMAP + ERASE | 23.34 | 5.91 |
| Implicatures | Random In-Context Sets | 29.67 | 11.55 |
| | ACoT | 29.76 | 2.95 |
| | ERASE | 32.37 | 14.11 |
| | UMAP + ACoT | 32.52 | 12.10 |
| | UMAP + ERASE | 28.21 | 8.05 |
| Intent Recognition | Random In-Context Sets | 55.89 | 13.62 |
| | ACoT | 77.17 | 0.60 |
| | ERASE | 57.76 | 12.73 |
| | UMAP + ACoT | 61.00 | 1.41 |
| | UMAP + ERASE | 64.63 | 12.05 |

implicit definition of exact unlearning—as any procedure that does not use a datapoint—falls into the class of algorithmic definition issues highlighted in Thudi et al. (2022b), where such flexibility can trivialize unlearning. In contrast, our work follows the more rigorous definition from Thudi et al. (2022b), which requires exact unlearning to be defined with respect to a specific training and exact unlearning algorithm.

*Table 6.* Pairwise p-values between algorithms (ACoT, ERASE) across tasks, testing the Null Hypothesis that $\mu_i = \mu_j$. We consider p-value less than 0.05 statistically significant and bold them.

| Task | Algorithm | ACoT | ERASE |
|---|---|---|---|
| Disambiguation QA | Random | **0.0000** | 0.384 |
| | ACoT | – | **0.0000** |
| Fantasy Reasoning | Random | 0.6436 | **0.0029** |
| | ACoT | – | **0.0005** |
| Implicatures | Random | 0.9695 | 0.4054 |
| | ACoT | – | 0.3133 |
| Intent Recognition | Random | **0.0000** | 0.5728 |
| | ACoT | – | **0.0000** |

*Table 7.* List of all training parameters used. The model used is described in Touvron et al. (2023)

| Parameter | Value |
|---|---|
| Trainable Parameters | All |
| Model | LLaMA 7B |
| Learning Rate | 1e-5 |
| Warm-up Steps | 10 |
| Adam Beta 1 | 0.9 |
| Adam Beta 2 | 0.98 |
| Weight Decay | 0.01 |
| Num Epochs | 1 |
| Eval Steps | 4 |
| Weight Precision | float 16 |
| Block Size | 256 |
| Num Micro Batches | 2 |
| Operator Parallel | 4 |
| Pipeline Parallel | 1 |

