# OpenReview forum: "Fast Exact Unlearning for In-Context Learning Data for LLMs"
_ICML.cc/2025/Conference — ICML 2025 poster_

### Official Review · Reviewer_ygwC · 2025-03-12

**Overall Recommendation:** 2

**Summary:**

This paper proposes ERASE, an in-context learning method combined with quantized k-means clustering for exact unlearning in LLMs. ERASE combines in-context learning with quantized k-means clustering, aiming to achieve dataset- and model-independent unlearning costs while maintaining competitive performance. A "holistic unlearning cost" metric is introduced to analyze trade-offs between inference and unlearning efficiency. The authors compare ERASE to SISA-based baselines on Big-Bench tasks, demonstrating reduced unlearning costs and comparable accuracy.

**Claims And Evidence:**

1. The paper claims dataset-independent unlearning costs based on quantized k-means, but fails to provide adequate justification for why this specific combination would be uniquely effective for LLMs. While quantized k-means is a known technique, its application to LLM embeddings lacks a foundation, particularly regarding stability in high-dimensional spaces.
2. The proposed metric C(M) for Holistic Unlearning Cost represents a ratio between unlearning and inference costs, but without proper empirical validation, and it lacks experimental measurements of actual computational costs (such as GPU time and memory usage) to substantiate its claims.

**Essential References Not Discussed:**

Although the paper cites Pawelczyk et al.'s work on in-context unlearning in the related work section, it fails to provide any meaningful discussion or comparison with this directly relevant approach.

[Pawelczyk et al.] Pawelczyk, M., Neel, S., and Lakkaraju, H. In-context unlearning: Language models as few shot unlearners. arXiv preprint arXiv:2310.07579, 2023.

**Experimental Designs Or Analyses:**

- The paper lacks a proper unlearning evaluation framework. Specifically, there is no clear distinction between forget sets and remaining sets, nor is there a comprehensive evaluation of both model utility preservation and forgetting ability.
- Regarding the forgetting ability, the authors do not employ standard unlearning metrics, such as membership inference attacks or data extraction leakage tests, which are crucial for validating privacy guarantees.

**Methods And Evaluation Criteria:**

- The methodology section suffers from insufficient comparisons. The authors did not compare their approach against alternative clustering methods (like hierarchical clustering), making it difficult to substantiate claims of method superiority.
- The choice of Big-Bench as an evaluation benchmark is questionable, as it was not designed for unlearning tasks.
- The paper lacks comparisons with existing in-context unlearning methods, particularly the work by Pawelczyk et al.

[Pawelczyk et al.] Pawelczyk, M., Neel, S., and Lakkaraju, H. In-context unlearning: Language models as few shot unlearners. arXiv preprint arXiv:2310.07579, 2023.

**Other Comments Or Suggestions:**

N/A

**Other Strengths And Weaknesses:**

N/A

**Questions For Authors:**

N/A

**Relation To Broader Scientific Literature:**

While most previous work focused primarily on model fine-tuning approaches for unlearning, this paper introduces a novel in-context learning perspective. Additionally, the paper attempts to address a critical gap in evaluation methodology by proposing a holistic unlearning cost framework.

**Theoretical Claims:**

N/A

---

> ### Author Rebuttal · Authors · 2025-03-31
>
> We thank the reviewer for their feedback and time. We respond to specific concerns below.
>
> > The paper claims dataset-independent unlearning costs based on quantized k-means, but fails to provide adequate justification for why this specific combination would be uniquely effective for LLMs. While quantized k-means is a known technique, its application to LLM embeddings lacks a foundation, particularly regarding stability in high-dimensional spaces.
>
> We wish to point out that our experiments showed Q-Kmeans was as effective as ACoT (which uses standard kmeans++) on the LLM eval datasets we tested, and also as effective as parameter fine-tuning with SISA. While in general this might not be true if we change the embedding dimension used in ERASE, a contribution of our paper is showing that for the tasks we tested (and when using BERT embeddings) it was effective.
>
> > The proposed metric C(M) for Holistic Unlearning Cost represents a ratio between unlearning and inference costs, but without proper empirical validation, and it lacks experimental measurements of actual computational costs (such as GPU time and memory usage) to substantiate its claims.
>
> Our experiments did measure actual computational costs. The results in Tables 3 and 4 (in the appendix) computed the FLOPS for training and inference, which are used to directly compare between methods (relative improvements are shown in Table 1).  We use FLOPS as opposed to GPU time and memory usage because FLOPS are independent of available hardware and, thus, more accurately capture algorithmic costs.
>
> > The methodology section suffers from insufficient comparisons. The authors did not compare their approach against alternative clustering methods (like hierarchical clustering), making it difficult to substantiate claims of method superiority.
>
> There are a limited number of efficient unlearnable clustering algorithms. We are unaware of efficient unlearning operations for hierarchical clustering.
>
> > The choice of Big-Bench as an evaluation benchmark is questionable, as it was not designed for unlearning tasks.
>
> We agree other datasets would be beneficial, but want to point out our evaluation was over 15 different tasks. These tasks already allowed us to conclude that which is better between in-context learning and fine-tuning is data dependent.
>
> > The paper lacks comparisons with existing in-context unlearning methods, particularly the work by Pawelczyk et al.
>
> In this paper we obtain exact unlearning guarantees, while the mentioned paper (also cited in our related work) only does approximate unlearning (and that too without any guarantees). It is currently difficult to comprehensively evaluate such approximate unlearning methods, e.g., [1], which further motivates our paper. We will make this clearer in our related works section.
>
> [1] Hayes, Jamie, et al. "Inexact unlearning needs more careful evaluations to avoid a false sense of privacy." arXiv preprint arXiv:2403.01218 (2024).
>
> > The paper lacks a proper unlearning evaluation framework. Specifically, there is no clear distinction between forget sets and remaining sets, nor is there a comprehensive evaluation of both model utility preservation and forgetting ability.
>
> To clarify, this distinction is only  present in approximate/heuristic unlearning where their effectiveness depends on the sets. Our method is an exact unlearning method, which by definition always works. Our evaluation thus focuses on exact unlearning compute.
>
> > Regarding the forgetting ability, the authors do not employ standard unlearning metrics, such as membership inference attacks or data extraction leakage tests, which are crucial for validating privacy guarantees.
>
> As mentioned earlier, our method is an exact unlearning method and so by definition will always have perfect unlearning performance under the approximate unlearning metrics. To elaborate, we are producing exactly the retrained models, and those metrics measure differences (of approximately unlearnt models) to the retrained model.
>
> > Although the paper cites Pawelczyk et al.'s work on in-context unlearning in the related work section, it fails to provide any meaningful discussion or comparison with this directly relevant approach.
>
> See our response to the earlier question about this paper; we will add more discussion elaborating the difference between their approximate unlearning approach and the guarantees of exact unlearning (which our method does) in our revised draft.

---

### Official Review · Reviewer_6e2H · 2025-03-13

**Overall Recommendation:** 2

**Summary:**

The paper proposes an exact unlearning algorithm, ERASE, for in-context learning. The core unlearning idea revolves around performing exact unlearning in AutoCOT, which clusters in-context examples using their sentence representations and uses the samples close to the cluster centroids as the ICL examples. ERASE leverages the quantized k-means algorithm to perform exact unlearning that efficiently updates the clustering process to remove deleted samples. The paper introduces a holistic evaluation strategy by accounting for both the unlearning and inference costs. The paper also provides several experiments on BigBench datasets comparing ERASE with baseline algorithms.

**Claims And Evidence:**

**Strengths**:

1. The paper is fairly easy to follow.

2. The paper introduces a neat paradigm for performing exact unlearning while using in-context learning. The paper proposes ERASE, an efficient exact unlearning algorithm to delete instances when using ACoT. The core idea is generalizable and can be utilized in any algorithm using clustering to select ICL examples.

3. The paper prioritizes an important aspect of unlearning, where unlearning and inference costs are often at odds. It introduces a metric that captures both costs, allowing for a holistic evaluation of an unlearning algorithm's performance.

4. Empirical evaluation shows that the proposed method, ERASE, achieves similar performance to baseline algorithms even after unlearning several instances.

**Weaknesses**:

1. The novelty of the technical contribution is weak. This is because the proposed approach uses the exact algorithm for unlearning used in clustering and applies it in ICL where clustering is involved. I understand that the algorithm has been used in a new application and other metrics for evaluating unlearning have been proposed. However, the technical contribution is weak because the core unlearning technique is borrowed from previous work and applied in a new setting.

2. Some of the underlying assumptions related to this work need to be mentioned. For example, for exact unlearning in this setup, the underlying assumption is that the ICL examples were not used during any stage of pre-training or instruction tuning.

3. The paper focuses on unlearning for a specific approach, ACoT. Although the paper motivates its advantage over vanilla CoT and showcases the results in Fig. 2 & 3, its significance over Random COT (random ICL example) selection is unclear in the context of unlearning. I understand that ACoT achieves better results than Random selection. But since Random CoT can perform O(1) deletion, the ideal way to evaluate would be to have a tradeoff between per query deletion cost and the overall performance (normalized aggregate score). In the current version, the paper reports only the normalized scores or inference cost at a time.

4.  The above point also raises a question about the holistic evaluation approach. Apart from inference and unlearning costs, shouldn’t we also consider the overall performance of the algorithm as a dimension to consider? For example, Random CoT has the same inference cost as ACoT and a much lower unlearning cost but the overall performance is poor compared to ACoT.

**Essential References Not Discussed:**

N/A

**Experimental Designs Or Analyses:**

N/A

**Methods And Evaluation Criteria:**

N/A

**Other Comments Or Suggestions:**

Line 102:  “While no past work studied unlearning specifically for the fine-tuning stage (i.e., task adaptation stage) of LLMs” — this claim isn’t accurate. There is this recent paper [1] that exclusively focuses on erasing fine-tuning data. Moreover, existing exact unlearning algorithms like SISA and others, can be modified to adapt to the fine-tuning setting.

[1] https://arxiv.org/pdf/2406.16257

**Other Strengths And Weaknesses:**

N/A

**Questions For Authors:**

Please respond to the weaknesses.

**Relation To Broader Scientific Literature:**

N/A

**Theoretical Claims:**

N/A

---

> ### Author Rebuttal · Authors · 2025-03-31
>
> We thank the reviewer for their time and feedback. We discuss specific points below.
>
> > Some of the underlying assumptions related to this work need to be mentioned. For example, for exact unlearning in this setup, the underlying assumption is that the ICL examples were not used during any stage of pre-training or instruction tuning.
>
> That is correct, and we will make that assumption clearer in our revised draft.
>
> > The paper focuses on unlearning for a specific approach, ACoT. Although the paper motivates its advantage over vanilla CoT and showcases the results in Fig. 2 & 3, its significance over Random COT (random ICL example) selection is unclear in the context of unlearning. I understand that ACoT achieves better results than Random selection. But since Random CoT can perform O(1) deletion, the ideal way to evaluate would be to have a tradeoff between per query deletion cost and the overall performance (normalized aggregate score). In the current version, the paper reports only the normalized scores or inference cost at a time.
>
> We would like to point out one more subtlety, which is ACoT and ERASE do not always perform better than random; this is dataset dependent. We decided to present solely the performance cost as it was already explained earlier in the paper that Random COT has O(1) cost, so it is up to the model provider to decide if the performance loss (on some datasets) is worth the cheaper unlearning cost. It is not clear to us how to value performance gains relative to unlearning computation. We will add this discussion elaborating the performance vs. unlearning consideration to our revised draft (in Section 5.2 if space allows).
>
> >The above point also raises a question about the holistic evaluation approach. Apart from inference and unlearning costs, shouldn’t we also consider the overall performance of the algorithm as a dimension to consider? For example, Random CoT has the same inference cost as ACoT and a much lower unlearning cost but the overall performance is poor compared to ACoT.
>
> See our response to the previous question, which provides a detailed discussion on the context dependent nature of the performance and unlearning comparison: it is not clear to us how to value performance with changes in compute. This was unlike inference and unlearning operations, which are both measured with the same computational units (e.g., flops).
>
> > Line 102: “While no past work studied unlearning specifically for the fine-tuning stage (i.e., task adaptation stage) of LLMs” — this claim isn’t accurate. There is this recent paper [1] that exclusively focuses on erasing fine-tuning data. Moreover, existing exact unlearning algorithms like SISA and others, can be modified to adapt to the fine-tuning setting.
>
>
> Thank you for bringing this paper to our attention! We will change our claim to mention this work, but note that it actually does not satisfy exact unlearning (according to our reading of their paper). Specifically, they propose to revert to a checkpoint that did not observe the datapoint to unlearn, but this does not reproduce the apriori trained model that would come from not having a datapoint in that slice. Furthermore, the selection of models amongst those trained from many permuted slices again means we do not reproduce the apriori distribution of models coming from not having a datapoint. This latter technique also seems analogous to the forging attack to fool exact unlearning [1]. More generally, the mentioned paper’s implicit definition of exact unlearning (which is anything that does not “use” a datapoint is exact unlearning) falls under the algorithmic definition issues pointed out in [1]; unlearning needs to be defined w.r.t a specific training/unlearning algorithm, as allowing anything that does not use a datapoint means we can trivially unlearn. After the submission cycle we will reach out to those authors to let them know of this issue.
>
> Note, the exact unlearning definition we follow removes this issue, by defining a specific training algorithm and unlearning with respect to it.
>
> [1] Thudi, Anvith, et al. "On the necessity of auditable algorithmic definitions for machine unlearning." 31st USENIX security symposium (USENIX Security 22). 2022.

---

### Official Review · Reviewer_s2NR · 2025-03-13

**Overall Recommendation:** 2

**Summary:**

This paper studies unlearning for in-context learning task adaptation, which is claimed to be understudied. Unlearning is mostly studied in setting where parameters are updates are required such as SISA. The overall method is a follow up on ACoT but in a new setting which is unlearning (ACoT was in learning paradigm) where the authors have used quantized kmeans++. The improvements is shown to efficiency of unlearning.

**Claims And Evidence:**

Some of the claims in the contribution are difficult to verify for instance.

1. The first claim in line 073-074 is vague and I am not sure how to verify it.

2. The claim 3 and 4 are also not novel and vague.

I request the authors to re-write the whole summary of contributions clearly.

**Essential References Not Discussed:**

None.

**Experimental Designs Or Analyses:**

The evaluation metrics are taken form prior works and they seems reasonable. Some of my concerns.

1. The comparison to SISA in sec 5.3 and 5.4 seems out of place as SISA is built for unlearning gradient based methods and ERASE is unlearning in-context finetuning.

2. Why the naive baseline of removing samples from context not reported?

3. The model and training details are very confusing to me. For instance which LLaMA?

4. I think the evaluation needs to be extended some harder tasks such as math generation gsm8K, MMLU , hellaswag and other metrics commonly used in LLM literature for fine-tuning.

**Methods And Evaluation Criteria:**

The authors need to explain well why unlearning during in-context learning (i.e. no change in parameters) is an important problem. A naive baseline in this setting in removing samples from incontext rather than analyzing the embeddings. The authors have explained this point from line 161-172, however this naive baseline is missing from results. The evaluation criteria is taken from prior works and seems reasonable.

A key limitation of this work based on my understanding is in-context fine-tuning doesn't work well for preference or instruction tuning like settings. So, ERASE in it's current form can't be applied to this important and widely used setting of fine-tuning.

**Other Comments Or Suggestions:**

Already discussed above.

**Other Strengths And Weaknesses:**

Strengths

1. This paper studies a new direction.

Weaknesses, apart from what already discussed.

1. The writing can be further improved.

2. The novel of the approach seems a bit limited as it directly extends ACoT [1] to unlearning. But since it is a fairly new area in my opinion this not a key concern.

**Questions For Authors:**

Do you believe ERASE can be extended for unlearning instruction samples from an instruction tuning dataset?

**Relation To Broader Scientific Literature:**

This paper extends ACoT [1] in unlearning setting and employs quantized Kmeans++ instead of regular kmeans++.



[1] Automatic chain of thought prompting in large language models, zhang et al. ICLR 2022.

**Theoretical Claims:**

no theoretical claims, I have verified that inference cost is correct in Table 1.

---

> ### Author Rebuttal · Authors · 2025-03-31
>
> We thank the reviewer for their feedback and time.
>
> > Some of the claims in the contribution are difficult to verify for instance. 1) The first claim in line 073-074 is vague and I am not sure how to verify it. 2) The claim 3 and 4 are also not novel and vague.I request the authors to re-write the whole summary of contributions clearly.
>
> Thank you for pointing this out, we propose to change the contributions bullet points as follows:
>
> 1) Showing that for certain datasets, exact unlearning can be efficient by using in-context learning.
> 2) Proposing an exact unlearning algorithm ERASE for in-context learning, that has dataset and model-size independent unlearning operation costs.
> 3) A study of when in-context learning provides more efficient unlearning deployments than fine-tuning (and repeating fine-tuning from scratch to unlearn), including changes to inference costs.
>
> > The authors need to explain well why unlearning during in-context learning (i.e. no change in parameters) is an important problem. A naive baseline in this setting in removing samples from incontext rather than analyzing the embeddings. The authors have explained this point from line 161-172, however this naive baseline is missing from results.
>
> To clarify, we understand the reviewer to be referring to random in-context selection, whose unlearning operation is replacing the selected example (which by rejection sampling one can see now samples from the correct unlearnt distribution). We have comparisons to this in Figure 2.
>
> We wish to also clarify that only removing the unlearnt in-context example is not a correct unlearning operation for any in-context learning algorithm. Specifically, unlearning is an algorithmic definition, and if the original selection used group statistics (e.g., clustering), we must simulate those decisions on the dataset without the example to unlearn. Motivating this, clustering based in-context learning has been observed to occasionally outperform random in-context selection, and a contribution of this paper is to note this is still efficient to unlearn.
>
> We will incorporate this previous paragraph into the related works to help motivate in-context unlearning.
>
> > A key limitation of this work based on my understanding is in-context fine-tuning doesn't work well for preference or instruction tuning like settings. So, ERASE in it's current form can't be applied to this important and widely used setting of fine-tuning.
>
> We will acknowledge this limitation in a revised draft. To be clear, the main claim of the paper is that for some fine-tuning datasets, exact unlearning can be easy as in-context learning can be as performant as fine-tuning.
>
> > The comparison to SISA in sec 5.3 and 5.4 seems out of place as SISA is built for unlearning gradient based methods and ERASE is unlearning in-context finetuning.
>
> The claim of our paper is that, when possible, we should do in-context learning instead of fine-tuning as it is much more efficient to unlearn. To show this we must compare in-context methods (ERASE) to their fine-tuning based counterparts (SISA). These sections show that in-context methods have comparable performance (on the datasets we tested) while dramatically reducing unlearning cost.
>
> > Why the naive baseline of removing samples from context not reported?
>
> It is in Figure 2, see our response to an earlier comment.
>
> > The model and training details are very confusing to me. For instance which LLaMA?
>
> As mentioned on line 267 we use the original LLaMA model from Touvron et al. (2023). The “Fine-Tuning Setup,” “Hyperparameter Selection,” and “Prompt Formatting” sub-sections lines 305-329 outline what we believe to be the most important training hyperparameters. In our revision we will improve readability by adding a table to our appendix showing a complete list of all training hyperparameters.
>
> >I think the evaluation needs to be extended some harder tasks such as math generation gsm8K, MMLU , hellaswag and other metrics commonly used in LLM literature for fine-tuning.
>
> Our existing evaluation includes 15 tasks capturing a variety of difficulties, and already allows us to conclude that which is better between in-context learning and finetuning is data dependent.
>
> > Do you believe ERASE can be extended for unlearning instruction samples from an instruction tuning dataset?
>
> We see no issue in using ERASE on other datasets where in-context learning is possible. However, the primary question is whether in-context learning is effective at learning from that dataset, and as pointed out by the reviewer, in-context learning tends to perform poorly on instruction tuning.

---

### Official Review · Reviewer_d3Vc · 2025-03-14

**Overall Recommendation:** 4

**Summary:**

this paper proposes a novel exact un-learning approach for in-context learning. Given a training sample, the goal of exact unlearning is to obtain as quickly as possible an algorithm that one would have obtained without training on that data sample. The authors study this problem in the context of in-context learning, where the algorithm is “trained” by seing several examples from the target dataset. Auto chain of thought is an efficient method for ICL, which clusters the examples into k clusters and only uses the k examples closest to the centroids for ICL. The authors propose instead to use a quantized k-means, which enables fast exact un-learning. The resulting method, ERASE, is then compared against other unlearning methods for ICL, like random selection (which performs worse than Acot but makes unlearning easier) and Acot (which requires re-running k-means at each unlearning step). The authors also compare their method to unlearning with standard SGD-based fine-tuning, SISA. The authors explore the results in the space of resulting inference cost and cost of unlearning.

# Edit after rebuttal + discussion

- This paper studies exact unlearning in the context of in-context learning. In-context learning is a very important topic of research in those days, as it is an efficient method to adapt models for some tasks (not all, of course). Therefore, developing un-learning algorithms for ICL is a very important research direction. This paper is the first paper to propose such algorithms for exact un-learning. With that in mind, I think that the main criticism of s2nr, that"*in-context fine-tuning doesn't work well for preference or instruction tuning like settings. So, ERASE in it's current form can't be applied to this important and widely used setting of fine-tuning*", is not valid criticism: ERASE is not supposed to work where ICL does not; it is clearly not this paper's purpose to make ICL work for these tasks. In my view, showing that the proposed method works in settings where ICL is beneficial is enough to validate the method.

- I also disagree with rev. 6e2H that there is little novelty: indeed, this paper is the first to propose exact un-learning algorithms for ICL. Unlearning and ICL are both important topics, and having even a baseline for unlearning ICL is important for the community as a whole. In my view, this paper fills an important gap. The fact that the method is simple is, in my view, a feature of the method and not a weakness. The method is novel and simple.

- I also found the response of the authors to rev. ygwC convincing; the bulk of this reviewer's criticism is about comparison to approximate un-learning methods; but comparing approximate and exact un-learning methods is notoriously hard to do; here the methods proposed by the authors provably un-learns (of course, this trades-off against efficiency compared to approximate unlearning, but again, in my view, this is not this paper's battle to benchmark approximate vs exact unlearning)

Therefore, I still think that this is an important article that should be accepted.

**Claims And Evidence:**

- ICL is a very popular approach, designing algorithms for exact unlearning on it is a novel and promising resarch direction
- the proposed method is sound, and the theory behind it is both simple to understand and elegant
- the empirical evaluation is quite thorough

- the main weakness to me is the title and positioning itself. It is the first time that I see ICL defined as a fine-tuning method. I think that for the vast majority of the community, fine-tuning means using a gradient based approach to tune or add parameters to the model, while ICL does not use gradient descent. I think that “fast exact unlearning for in-context learning data for LLMs” or something along those lines would make what the paper is about much clearer.

**Essential References Not Discussed:**

- the paper "In-context unlearning: Language models as few shot unlearners.” seems related, it would be good to clarify more the differences with the present approach.

**Experimental Designs Or Analyses:**

- the error bars in most experiments are huge, it would be good to have a sense of the statistical significance of the results of the paper.

**Methods And Evaluation Criteria:**

- the proposed method is sound, and the theory behind it is both simple to understand and elegant

- gradient based fine-tuning and ICL have very different effects on the model. gradient based fine-tuning scales much better when the dataset size increases (and is worse at small scales), and it incurs no inference overhead. I think that the experiments could do a better job at exploring this space.
- the error bars in most experiments are huge, it would be good to have a sense of the statistical significance of the results of the paper.

**Other Comments Or Suggestions:**

- "this, we have unlearning the fine-tuning data is independent of the model and dataset size” unclear to me
- “Showing that learning with access to an LLM can allow for faster exact unlearning.” unclear to me
- “makes undoing x fast” unlearning?
- “our models converge after one epoch of SISA finetuning.” isn’t it contrary to the conventional wisdom that repeating data a few epochs improves performance?
- It would be good to clarify the costs of inference for ICL. The fact that it is linear with the number of token puzzles me, my understanding is that attention is quadratic in the number of tokens. One can use caching to make the cost linear with the ICL prompt length, but that cost still seems quadratic the first time it is done.
- I would use in fig 3 a different color palette than that in fig 2

**Other Strengths And Weaknesses:**

- the paper is very easy to follow, it was a joy to read

**Questions For Authors:**

- would it be possible to extend results in table 2 in a 2d place, where x axis is unlearning cost and y axis is inference cost?
- In fig 2, how can such a slight modification of the algorithm yield such changes ? can we conclude anything based on this, looking at the error bars?
- How does the method scale with dataset size? I expect that at large datasets sizes, ICL starts lagging behind fine tuning.

**Relation To Broader Scientific Literature:**

- the paper "In-context unlearning: Language models as few shot unlearners.” seems related, it would be good to clarify more the differences with the present approach.

**Theoretical Claims:**

- the proposed method is sound, and the theory behind it is both simple to understand and elegant

---

> ### Author Rebuttal · Authors · 2025-03-31
>
> We thank the reviewer for their time and feedback! Below we discuss questions raised in the review. Other suggestions will be implemented in our revised draft.
>
> >the main weakness to me is the title and positioning itself. It is the first time that I see ICL defined as a fine-tuning method...
>
> We agree that the terminology in-context learning is more clear than fine tuning and propose to change the title (“fast exact unlearning for in-context learning data for LLMs” sounds good to us) and contributions accordingly. See our response to Reviewer s2NR for specific changes to the contribution bullet points. To clarify, this does not change any of the methodology or experimental setup and the results hold.
>
> > the error bars in most experiments are huge, it would be good to have a sense of the statistical significance of the results of the paper.
>
> While the error bars are large in terms of accuracy, the methods perform differently (in a statistically meaningful way) when it comes to unlearning cost. That is, the FLOPS required to unlearn varies between the methods, despite their accuracy being comparable.
>
> > the paper "In-context unlearning: Language models as few shot unlearners.” seems related, it would be good to clarify more the differences with the present approach.
>
> In this paper we obtain exact unlearning guarantees, while the mentioned paper (also cited in our related work) only does approximate unlearning (and that too without any guarantees). It is currently difficult to comprehensively evaluate such approximate unlearning methods, e.g., [1], which further motivates our paper.
>
> [1] Hayes, Jamie, et al. "Inexact unlearning needs more careful evaluations to avoid a false sense of privacy."
>
> > “our models converge after one epoch of SISA finetuning.” isn’t it contrary to the conventional wisdom that repeating data a few epochs improves performance?
>
> It is common with LLMs to only train for one epoch. Supporting this, we generally found that the training loss for our tasks reached near 0 within a single epoch; we hence stop at one epoch to avoid overfitting. We will add a note about this in the appendix with associated training loss graphs.
>
> > It would be good to clarify the costs of inference for ICL. The fact that it is linear with the number of token puzzles me, my understanding is that attention is quadratic in the number of tokens. One can use caching to make the cost linear with the ICL prompt length, but that cost still seems quadratic the first time it is done.
>
> Thank you for pointing this out. You are correct that the asymptotic relationship should be quadratic in terms of number of tokens in the context. The asymptotic costs in table 1 will be updated in the revision with “t” -> “t^2”. We note that this has a negligible effect on our experimental results as we manually measured inference flops. The primary effect is improving ERASE’s relative asymptotic cost efficiency against the baseline SISA method.
>
> > In fig 2, how can such a slight modification of the algorithm yield such changes ? can we conclude anything based on this, looking at the error bars?
>
> We were unsure which “slight modification” you are referring to. We will list our response to a couple of modifications:
>
> Random vs ACoT: Our results suggest that ACoT performance benefits are task dependant. The ACoT paper also states that it “matches or exceeds the performance of the CoT paradigm that requires manual designs of demonstrations." However, we believe further work is required to accurately understand this dependance.
>
> UMAP vs non-UMAP: The dimension of our text embeddings is large (1536). It is a known that clustering may work poorly in high dimensional settings. Thus, applying dimensionality reduction to our text embeddings could have potentially improved clustering results.
>
> ACoT vs ERASE: We find that in-context learning performance is very sensitive to which examples are chosen, causing high variance for each method. We found on average ERASE and ACoT perform similarly, and believe a significant portion of the variance is intrinsic to the choice of in-context examples.
>
> > How does the method scale with dataset size? I expect that at large datasets sizes, ICL starts lagging behind fine tuning.
>
> We agree that if there is an abundance of data, and the task is sufficiently different from the pre-trained model’s data, fine-tuning will perform better. However, this comes with additional unlearning costs. We will make it clearer in the paper that our focus is on the cases where in-context learning performs similarly to fine-tuning.

---

> > ### Comment · Reviewer_d3Vc · 2025-04-03
> >
> > Dear authors,
> >
> > Thanks for the clarification.
> >
> > > We agree that the terminology in-context learning is more clear than fine tuning and propose to change the title (“fast exact unlearning for in-context learning data for LLMs” sounds good to us) and contributions accordingly. See our response to Reviewer s2NR for specific changes to the contribution bullet points. To clarify, this does not change any of the methodology or experimental setup and the results hold.
> >
> > I think that this will indeed make the point of the paper clearer.
> >
> > > While the error bars are large in terms of accuracy, the methods perform differently (in a statistically meaningful way) when it comes to unlearning cost. That is, the FLOPS required to unlearn varies between the methods, despite their accuracy being comparable.
> >
> > I agree that the computational gains are meaningful. However, there are a few places in the manuscript where you claim that a method is better than another despite large error bars (e.g. *"We see that ERASE matches our outperforms ACoT on three of the four tasks, and similarly with random selection. Considering dimension reduction for ERASE , we observed it made slight improvements "*). These claims are void if they do not come with a statistical test, please report the p values you get with that experiment and update the caption accordingly.
> >
> > > In this paper we obtain exact unlearning guarantees, while the mentioned paper (also cited in our related work) only does approximate unlearning
> >
> > Thanks for the clarification
> >
> > > It is common with LLMs to only train for one epoch. Supporting this, we generally found that the training loss for our tasks reached near 0 within a single epoch
> >
> > These two sentences do not seem related to me; indeed, LLMs are pretrained on non-repeated data, but the loss is far from 0. The fact that the training loss is close to 0 within a single epoch is very surprising; how can the model memorize each data point with a single gradient step per mini-batch? Conventionnal wisdom when training LLMs is that data can be repeated ~3 times without hurting the generalization performances (see [1]). This casts some doubts on the practical implementation of SISA.
> >
> >
> > > Thank you for pointing this out. You are correct that the asymptotic relationship should be quadratic in terms of number of tokens in the context. The asymptotic costs in table 1 will be updated in the revision with “t” -> “t^2”. We note that this has a negligible effect on our experimental results as we manually measured inference flops. The primary effect is improving ERASE’s relative asymptotic cost efficiency against the baseline SISA method.
> >
> > Thanks for the clarification.
> >
> > > We were unsure which “slight modification” you are referring to. We will list our response to a couple of modifications:
> >
> > I apologize for not being clear about this; I was referring to UMAP vs non-UMAP. Thanks for the clarification.
> >
> > > We agree that if there is an abundance of data, and the task is sufficiently different from the pre-trained model’s data, fine-tuning will perform better. However, this comes with additional unlearning costs. We will make it clearer in the paper that our focus is on the cases where in-context learning performs similarly to fine-tuning.
> >
> > Thanks; I understand it might be a lot of work, but an ablation regarding dataset size would help in having a broader picture.
> >
> > [1]:Muennighoff, Niklas, et al. "Scaling data-constrained language models." Advances in Neural Information Processing Systems 36 (2023): 50358-50376.

---

> > > ### Author Response · Authors · 2025-04-03
> > >
> > > Thank you for your response!
> > >
> > > > I agree that the computational gains are meaningful. However, there are a few places in the manuscript where you claim that a method is better than another despite large error bars (e.g. "We see that ERASE matches our outperforms ACoT on three of the four tasks, and similarly with random selection. Considering dimension reduction for ERASE , we observed it made slight improvements "). These claims are void if they do not come with a statistical test, please report the p values you get with that experiment and update the caption accordingly.
> > >
> > > Yes we completely agree, we will include p values and adjust the claims in the paper accordingly. Below is a table containing the pairwise p-values between algorithms (ACoT, ERASE) across tasks, testing the null hypothesis that the means of two methods (for a given dataset) were the same. This is for the experiments in Figure 2 with no UMAP. We found most of the comparisons to be significant, and bolded those with p values $< 0.05$.
> > >
> > >
> > > | Task                | Algorithm | ACoT         | ERASE        |
> > > |---------------------|-----------|--------------|--------------|
> > > | **Disambiguation QA** | Random    | **0.0000**   | 0.384        |
> > > |                     | ACoT      | --           | **0.0000**   |
> > > | **Fantasy Reasoning** | Random    | 0.6436       | **0.0029**   |
> > > |                     | ACoT      | --           | **0.0005**   |
> > > | **Implicatures**      | Random    | 0.9695       | 0.4054       |
> > > |                     | ACoT      | --           | 0.3133       |
> > > | **Intent Recognition** | Random    | **0.0000**   | 0.5728       |
> > > |                     | ACoT      | --           | **0.0000**   |
> > >
> > >  > Thanks; I understand it might be a lot of work, but an ablation regarding dataset size would help in having a broader picture.
> > >
> > > Yes we agree, we currently don't believe we have the resources to run it in time for the rebuttal, but will at the very least acknowledge this limitation in our revised draft.
> > >
> > > > These two sentences do not seem related to me; indeed, LLMs are pretrained on non-repeated data, but the loss is far from 0. The fact that the training loss is close to 0 within a single epoch is very surprising; how can the model memorize each data point with a single gradient step per mini-batch? Conventionnal wisdom when training LLMs is that data can be repeated ~3 times without hurting the generalization performances (see [1]). This casts some doubts on the practical implementation of SISA.
> > >
> > > We think the difference is we are fine-tuning a pre-trained model, versus training a pre-trained model from scratch. We believe the fact just an epoch suffices to get near 0 training loss suggests the pre-trained model was already quite good for our task/ the fine-tuning tasks are relatively easy to learn compared to pre-training the LLM. We will make this distinction to pre-training clear in our revised draft.

---

### Decision · Program_Chairs · 2025-05-01

**Decision:**

Accept (poster)

**Comment:**

### Summary

This paper proposes a new unlearning method for in-context learning (ICL), coined ERASE, inspired by ACOT, in which the Kmeans++ is replaced by quantized Kmeans++. In particular, the paper focuses on finetuning/in-context learning settings, but **not** the pretraining stage. ERASE cost is independent of dataset size, unlike n-SISA and ACOT, thanks to the stability of quantized k-means. Another innovation of the paper is to take into account the inference cost of every method and balancing it with the cost of unlearning, and to estimate the number of inferences a method can handle before being outperformed by

### Discussions and rebuttal

Reviewer **ygwC** asked for empirical evaluation of the forgetting (e.g through membership attacks and practical benchmarks); while these additional experiments could strengthen by comparing against empirical methods, I don't think they are required *per-se* since the method is sound and already provide formal unlearning guarantees - therefore empirical evaluations would only be an ablation.

Reviewer **d3Vc** noticed that many empirical results were reported as improvement over competitors, but the score difference is actually much smaller than the error bars. Formal statistical test does not allow to conclude that ERASE dominates ACOT and the Random baseline. As it stands, ERASE is an algorithmic improvement regarding unlearning cost, but cannot be considered improvement regarding final utility over these baselines.

Reviewer **6e2H** found the contribution incremental, and encouraged the author to clarify that in the ICL setup, the forgetting of pretraining examples is not considered. Furthermore, a graph showcasing the tradeoff between forgetting cost and final performance seems necessary (for example a joint plot?), to show that the methods live on a Pareto front (which might differ from one task to another).

### Recommendation

ERASE is a simple yet effective improvement of ACOT, by replacing Kmeans++ with a quantized Kmeans++ it yields better inference/unlearning cost tradeoff with no negative impact on utility. However, unlike what the author claim, it cannot be said that ERASE outperforms ACOT, n-SISA or Random baseline based on the benchmark: the error-bars are too high to conclude a meaningful advantage - on some benchmarks it cannot even allow to conclude that any of these method perform better than random guessing (score = 0).

However, another contribution of the paper beyond ERASE is the inference/unlearning tradeoff analysis, which is an important direction for the community and a clear criterion on which methods should compare. This contribution is valuable in itself.

Therefore, I recommend acceptance. Nonetheless, I encourage the author to clarify the claims regarding efficiency of ERASE, to report statistical tests on every benchmark, and to clarify their unlearning setup.